# The Mediating Effects of Symptom Experiences on the Relationship between Body Image and Quality of Life among Hemodialysis Patients in a Single Center

**DOI:** 10.3390/healthcare12171779

**Published:** 2024-09-05

**Authors:** Yaki Yang

**Affiliations:** Department of Nursing, Wonkwang University, Iksan-si 54538, Republic of Korea; ykyang@wku.ac.kr; Tel.: +82-63-850-6019

**Keywords:** hemodialysis, body image, symptom experiences, quality of life

## Abstract

This study aimed to confirm the mediating effects of symptom experiences on the relationship between the body image and quality of life of hemodialysis patients. Data were collected from 153 patients who were diagnosed with ESRD at a tertiary general hospital and receiving regular hemodialysis in Korea. The data were collected between 20 July and 11 August 2023. The following statistical analyses were conducted: t-test, ANOVA, Scheffé test, Pearson’s correlation coefficient analysis, and Hayes’ Process Macro Model 4 (to test the mediating effect). The factors influencing quality of life were body image (β = 0.46, *p* < 0.001), monthly family income (over KRW 3 million) (β = 0.22, *p* = 0.002), and symptom experiences (β = −0.20, *p* = 0.001). The mediation analysis indicated that the symptom experiences mediated the relationship between body image and quality of life. Based on the results, education, counseling, and symptom management programs that can improve body image and reduce symptom experience should be developed, and customized programs that reflect the characteristics of a target population, such as economic level, should be developed and provided.

## 1. Introduction

The prevalence of chronic diseases, which are a major cause of death and disability worldwide, is increasing due to advances in medical technology, extension of lifespan, and changes in lifestyle habits, which is adding to the burden on individuals and society. Among the high-prevalence chronic conditions is Chronic Kidney Disease (CKD), characterized by progressive loss of kidney function over time. End-Stage Renal Disease (ESRD) represents the permanent final stage of CKD, where kidney function is severely impaired, posing a critical threat to the patient’s life.

Among end-stage renal disease patients receiving renal replacement therapy, hemodialysis accounts for the largest proportion with 107,015 individuals (79.4%), followed by kidney transplantation with 22,224 individuals (16.5%), and peritoneal dialysis with 5587 individuals (4.1%) in Korea [1]. Even overseas, the prevalence of patients with end-stage renal disease and dialysis is increasing [2,3,4], and in 2020, hemodialysis patients in Middle Eastern countries accounted for 75.8% of renal replacement therapy patients, and in Europe, 77.0% [4]. In the United States in 2023, 14% of US adults, that is, about 35.5 million people, were estimated to have chronic kidney disease, with 71% of them undergoing dialysis [5].

Hemodialysis is a method in which a dialysis machine replaces kidney function. Hemodialysis patients must live their entire lives dependent on dialysis treatment, which removes excess moisture and waste products accumulated in the body and maintains electrolyte balance in the body two to three times a week, for more than 4 h each time [6]. Even during the process of maintaining dialysis, they experience integrated physical, mental, and social problems, resulting in low quality of life [7] and a high risk of death and hospitalization [8]. It is different from other chronic diseases in that it extends life by relying on a dialysis machine [9], and the quality of life of patients is lower than that of the general public, peritoneal dialysis patients, and kidney transplant patients due to complex physical symptoms, frustration, depression, anxiety, and limitations in daily life [10]. Therefore, improving quality of life rather than merely extending lifespan is paramount for hemodialysis patients [11].

During hemodialysis treatment, patients often experience changes in appetite and weight [12]. Additionally, the process of dialysis results in loss of nutrients, muscle wasting due to protein catabolism, and decreased physical activity, leading to alterations in body appearance [11,13]. Furthermore, changes of body image can occur throughout the entire body due to endocrine disorders, appetite changes, edema, weight fluctuations, decreased libido, reduced muscle tone, surgical scars, changes in skin color, and arteriovenous fistula [14,15].

Research on the body image of hemodialysis patients reveals that 56.7% of them experience discomfort due to changes in body shape [16]. It has been reported that body image disturbances are significantly related to psychological distress [17]. Comparative studies between hemodialysis patients and kidney transplant recipients indicate a significantly higher prevalence of body image disturbances among hemodialysis patients [18]. Comparative studies with the general population also confirm markedly higher levels of body image disturbances among hemodialysis patients [17].

The changes of body image in hemodialysis patients influence their clothing choices [19]. Additionally, factors such as arteriovenous fistula, frequent scars from dialysis needles, and vascular enlargement due to aneurysm formation have a negative impact on sexual function [20]. Body image disturbances among hemodialysis patients lead to social anxiety, embarrassment, depression, decreased self-efficacy, and reduced quality of life [7]. It is difficult to engage in social and work life as before dialysis, and people experience problems such as depression and decreased quality of life [21,22].

Despite advancements in technology improving the performance of dialysis machines, hemodialysis remains a conservative treatment method aimed at periodically removing metabolic waste and excess fluid from the body to alleviate symptoms [6]. Even with proper hemodialysis, patients commonly experience physical symptoms such as dry skin, itching, muscle cramps, and fatigue, as well as emotional symptoms including decreased libido, sleep disorders, and irritability [7].

Hemodialysis patients typically experience an average of 5.7 to 7.5 interconnected symptoms [23], and their quality of life is influenced by the frequency and severity of these symptoms [24]. It has been observed that lower quality of life correlates with higher symptom severity, discomfort, and frequency of symptom experiences [25]. The treatment goal for hemodialysis patients is to adhere to treatment regimens during dialysis, manage chronic stress effectively, and maintain a good quality of life [26]. For hemodialysis patients, quality of life serves as a crucial indicator of adaptation to the changed lifestyle post-diagnosis and is also closely related to mortality rates [27].

Studies examining factors related to quality of life in hemodialysis patients have identified several key aspects. These include experiences of symptoms related to the disease [11,13], sleep disorders and restless legs syndrome [28], and comorbid conditions [29]. Psychological factors related to quality of life include depression [30], self-efficacy (Lee and Kim, 2015) [31], and resilience [25]. Environmental factors such as social support have also been identified [21]. Furthermore, research has focused on self-care behaviors and patient role performance to promote health behaviors among hemodialysis patients [32], education for accurate knowledge acquisition related to dialysis and disease management [33], as well as interventions involving diet [34] and exercise [35]. These studies underscore the multifaceted approach required to enhance the quality of life of hemodialysis patients, addressing both physical symptoms and psychological well-being, while also emphasizing the importance of education, self-care behaviors, and social support in their care.

However, research on the body image aspects related to hemodialysis patients has been insufficient, and studies specifically examining the relationship between body image, symptom experiences during dialysis, and quality of life in these patients are lacking. Existing studies have identified body image [14,15,18] and various symptom experiences during dialysis [7,11] as key factors influencing quality of life in hemodialysis patients. However, there remains a gap in understanding the specific mechanisms by which symptom experiences mediate the relationship between body image and quality of life. While preliminary studies have separately analyzed relationships of body image, symptom experiences, and quality of life among hemodialysis patients, there is a limitation in understanding the precise mediating role of symptom experiences in the relationship between body image and quality of life.

Therefore, this study aimed to assess the level of body image, symptom experiences, and quality of life among hemodialysis patients. Specifically, we sought to understand the mediating effects of symptom experiences in the relationship between body image and quality of life. Through this research, we aim to provide foundational data for the development of interventions aimed at improving the quality of life of hemodialysis patients. The four hypotheses of this study were formulated as follows:

**H1.** *Body image affects quality of life among hemodialysis patients*.

**H2.** *Body image affects symptom experiences among hemodialysis patients*.

**H3.** *Symptom experiences affect quality of life among hemodialysis patients*.

**H4.** *Symptom experiences are a mediating influence between body image and quality of life among hemodialysis patients*.

## 2. Materials and Methods

### 2.1. Study Design and Sample

This study was an instance of descriptive exploratory research aimed at examining the mediating effects of symptom experiences in the relationship between body image and quality of life among hemodialysis patients. This study targeted 153 patients diagnosed with end-stage renal disease receiving hemodialysis treatment at artificial kidney units located in J province in Korea. The participants were aged 19 years or older and had undergone hemodialysis therapy via temporary or permanent vascular access for at least 3 months. They understood the purpose and methods of this study and voluntarily agreed to participate. Patients currently exhibiting acute symptoms, undergoing severe illness requiring hospitalization, or diagnosed with Alzheimer’s or vascular dementia, terminal cancer, or psychiatric disorders were excluded from this study. The sample size was determined using G*power 3.1 software [36] for multiple regression analysis, requiring 143 participants to achieve a medium effect size of 0.15, a power of 0.80, and 16 predictor variables. Considering dropout rates, 160 participants were initially enrolled, and after excluding 7 incomplete questionnaires, data from 153 participants were included in the final analysis.

### 2.2. Instrument

A structured self-administered questionnaire was utilized, comprising a total of 139 items categorized into 14 items on general characteristics, 69 items on body image, 30 items on symptom experiences, and 26 items on quality of life. Before using the research instruments, approval for their use was obtained.

#### 2.2.1. Body Image

For the measurement of body image, the MBSRQ (The Multidimensional Body–Self Relations Questionnaire), developed by Cash [37] and adapted by Kang [38], was purchased through the authorized website (http://www.body-images.com) (accessed on 15 July 2023) and utilized. The MBSRQ is a self-report tool that evaluates body image from a self-perception perspective, consisting of 69 items. The MBSRQ measures the physical self across emotional and cognitive–behavioral dimensions. The emotional dimension consists of 5 subscales: appearance evaluation, body evaluation, health evaluation, satisfaction with specific body parts, and weight evaluation. Higher scores indicate greater positivity in these aspects. The cognitive–behavioral dimension comprises 5 subscales: appearance orientation, body orientation, health orientation, illness orientation, and obesity sensitivity. Higher scores indicate greater interest and proactive behavior regarding one’s appearance, body, health, illness, and sensitivity to obesity. Each subscale consists of the following items: 7 items for appearance evaluation, 3 items for body evaluation, 6 items for health evaluation, 9 items for satisfaction with specific body parts, 2 items for weight evaluation, 12 items for appearance orientation, 13 items for body orientation, 8 items for health orientation, 5 items for illness orientation, and 4 items for obesity sensitivity. Each subscale is measured on a Likert scale from 1 (“Not at all true”) to 5 (“Very true”), with higher scores indicating a more positive body image. In Kang’s study [38], Cronbach’s α ranged from 0.49 to 0.89 for each subscale, while in this study, Cronbach’s α was 0.88.

#### 2.2.2. Symptom Experiences

For the measurement of symptom experiences, the DSI (Dialysis Symptom Index) developed by Weisbord et al. [39] to assess both physical and emotional symptoms in hemodialysis patients was utilized. The tool was translated and adapted by Lim [40]. The instrument consists of two subdomains: physical symptoms with 21 items and emotional symptoms with 9 items. In the past week, symptoms were assessed using a Likert scale where the absence of symptoms was rated as ‘no symptoms’ with 0 points. For those experiencing symptoms, the intensity was rated from ‘not at all distressing’ with 1 point to ‘extremely distressing’ with 5 points. Higher scores indicate greater symptom intensity. In Lim’s study [40], Cronbach’s α was 0.90, and in this study, Cronbach’s α was also 0.90.

#### 2.2.3. Quality of Life

For the measurement of quality of life, the Korean version of the World Health Organization Quality of Life Scale (WHOQOL-BREF), revised and adapted by Min et al. [41], was used. This tool consists of 5 subdomains: overall quality of life (2 items), physical health domain (7 items), psychological health domain (6 items), social relationships domain (3 items), and environmental domain (8 items). The scale measures from “not at all” 1 point to “very much” 5 points on a Likert scale, where higher scores indicate a higher quality of life. In Min et al.‘s study [41], Cronbach’s α was 0.90, and in this study, Cronbach’s α was 0.88.

### 2.3. Data Collection

Data collection for this study was conducted at hospitals located in J Province from 20 July 2023 to 11 August 2023. Prior to data collection, permission was obtained from the hospital nursing department after explaining the purpose, participants, and methods of this study. Participants were recruited voluntarily through recruitment notices. For those capable of self-reporting, they were asked to complete the questionnaire independently. For participants with impaired vision or difficulty in writing, the researcher read the questionnaire aloud and recorded their responses. The average time required to complete the questionnaire was approximately 10 to 15 min.

### 2.4. Ethical Consideration

This study was conducted after obtaining approval from the Institutional Review Board (IRB) of the hospital (Approval Number: 2023-05-020-001) prior to data collection. Participants were informed about the purpose, methods, expected outcomes, and assurances of anonymity of the study to protect their personal information. It was clarified that their data would not be used for any purposes other than research. Participants were also provided with an explanation sheet stating that there would be no disadvantages for non-participation or withdrawal from the study. The questionnaires were collected by the researchers directly and were anonymously coded. The completed questionnaires were stored in a secure cabinet. After the conclusion of this study, the questionnaires will be kept for 3 years and then securely disposed of.

### 2.5. Statistical Analysis

Data analysis was conducted using SPSS/WIN 28.0 [42] and the PROCESS macro 3.0 (http://www.processmarco.org/index.html) (accessed on 15 September 2023). Descriptive statistics were used to analyze the levels of body image, symptom experience, and quality of life among participants, while Cronbach’s α was utilized to assess the reliability of the instruments. Differences in quality of life based on participants’ general characteristics were analyzed using independent *t*-tests and one-way ANOVA, with post hoc tests conducted using the Scheffé method. The correlations between body image, symptom experience, and quality of life were examined using Pearson’s correlation coefficient. To explore the mediating role of symptom experience in the relationship between body image and quality of life, SPSS PROCESS macro Model 4, as outlined by Hayes [43], was employed. Bootstrap methods were used to assess the statistical significance of the mediating effects.

## 3. Results

### 3.1. Participant Characteristics and Differences in Quality of Life

This study included a total of 153 participants, with males comprising 52.3%, and the majority of participants (37.3%) being in their 60s. Regarding marital status, 66.7% were married, and 52.9% reported having no religious affiliation. The unemployment rate was 84.3%, and the most monthly family income was less than KRW 1 million, accounting for 41.2%. Spouses were the primary caregivers for 45.1% of participants. The most common etiology of CKD was diabetes mellitus (58.2%), and 61.4% reported having comorbidities. The majority (30.1%) had been undergoing hemodialysis for 2~4 years.

Differences in quality of life according to general characteristics were observed in relation to religion, occupation, monthly family income, and the presence of other diseases besides renal failure. According to the Scheffé post hoc test results, monthly family incomes of KRW 2–2.99 million and KRW 3 million or more were significantly higher than those under KRW 1 million (Table 1).

### 3.2. Degrees of Body Image, Symptom Experiences and Quality of Life

The average score for body image was 2.88 ± 0.36 (out of 5 points). The emotional dimension of body image was rated at 2.67 ± 0.50 and the cognitive–behavioral dimension was rated at 3.02 ± 0.35. The average score for symptom experiences was 0.96 ± 0.79 (ranging from 0 to 5 points). Physical symptom experience was 0.94 ± 0.79, and emotional symptom experience was 1.03 ± 0.98, indicating higher emotional symptoms compared to physical symptoms. The average score for quality of life was 2.93 ± 0.46 (out of 5 points). The environmental domain had the highest score at 3.19 ± 0.59, while the overall quality of life domain had the lowest score at 2.55 ± 0.86 (Table 2).

### 3.3. Correlations among Body Image, Symptom Experiences, and Quality of Life

Quality of life showed a positive correlation with body image (r = 0.61, *p* < 0.001) (moderate) and a negative correlation with symptom experiences (r = −0.31, *p* < 0.001) (weak). Body image and symptom experiences were negatively correlated (r = −0.18, *p* = 0.023) (weak). In other words, higher body image scores were associated with higher quality of life, while higher symptom experiences scores were associated with lower body image and lower quality of life (Table 3).

### 3.4. Mediating Effect of Symptom Experiences between Body Image and Quality of Life

In order to verify the mediating effect of symptom experiences in the relationship between body image and quality of life among the subjects, the mediation models were examined as shown in Table 4 and Table 5, and Figure 1.

Regarding multicollinearity among the independent variables for the PROCESS macro analysis, the tolerance limits were all below 1.0, and the Durbin–Watson statistic for autocorrelation concerning quality of life was 2.05, indicating no significant autocorrelation (Durbin–Watson = 1.818). The Variance Inflation Factor (VIF) ranged from 1.042 to 1.446, all well below 10, indicating no multicollinearity among the independent variables.

In the multiple regression analysis assessing the impact on quality of life, monthly family income ranges of KRW 2–2.99 million (*p* = 0.049), and KRW 3 million or more (*p* = 0.002), along with body image (*p* < 0.001) and symptom experiences (*p* = 0.003), were found to significantly influence quality of life. Specifically, body image (β = 0.468), KRW 3 million or more (β = 0.223), symptom experiences (β = −0.187), and KRW 2–2.99 million (β = 0.129) showed varying degrees of influence on quality of life, with the model explaining 46.0% of the variance. Thus, higher body image scores were associated with higher quality of life, while incomes of KRW 2–2.99 million and KRW 3 million or more were associated with higher quality of life compared to incomes below KRW 1–1.99 million. Additionally, higher symptom experiences were associated with lower quality of life.

The mediating effect was tested using Hayes’ PROCESS macro Model 4, employing bootstrapping with 10,000 resamples and estimating bias-corrected 95% confidence intervals (CI). Bootstrapping reduces Type I errors associated with Baron and Kenny’s method and addresses errors related to the normality assumption of indirect effect distributions in Sobel tests. Controlling for occupation status, income, and diseases other than renal disease, the analysis revealed that body image (H1: B = 0.65, *p* < 0.001) and symptom experience (H3: B = −0.01, *p* = 0.003) significantly influenced quality of life. The PROCESS macro’s bootstrapping results for the mediating effect showed that quality of life mediated the impact of body image on symptom experiences (H4: B = 0.04, 95% CI: 006 to 0.100) that did not include zero, indicating a significant mediating effect. Therefore, it was confirmed that higher body image scores lead to lower symptom experiences, thereby increasing quality of life.

## 4. Discussion

This study aimed to assess the levels of body image, symptom experiences, and quality of life among hemodialysis patients, and to investigate the mediating effect of symptom experiences in the relationship between body image and quality of life.

Based on the study results examining the levels of body image, symptom experience, and quality of life among hemodialysis patients, the average score for body image was 2.88 out of 5 points. When examined by subdomains, the emotional dimension of body image was rated at 2.67, while the cognitive–behavioral dimension was rated at 3.02, indicating that the emotional dimension of body image was lower. These findings are consistent with previous research using the same instrument to measure body image in hemodialysis patients [44], although they are lower compared to studies measuring body image in breast cancer survivors [45]. The results supported previous research indicating that body image in hemodialysis patients is lower compared to the general population or individuals with other health conditions [44].

When evaluating images related to body image, the face occupies a significant portion. In hemodialysis patients, facial changes can be prominent due to hyperpigmentation, weight loss, repeated edema, and skin aging caused by treatment sessions. Additionally, frequent needle punctures for dialysis, formation of arteriovenous fistulas, and the presence of dialysis catheters contribute to visible bodily changes that cannot be easily concealed under clothing [15].

It is crucial for hemodialysis unit nurses to educate and prepare patients in advance about these bodily changes. Providing visual materials and educational pamphlets on body image changes can help patients adapt to the changes they may experience before starting dialysis.

Symptom experiences among hemodialysis patients averaged 0.96 points on a scale of 0 to 5, with physical symptom experience at 0.94 points and emotional symptom experience at 1.03 points, indicating that emotional symptoms were slightly higher than physical symptoms. This finding is similar to previous studies using the same instrument to assess symptom experience in hemodialysis patients [13]. However, the average score below 1 point in this study suggests that ongoing dialysis treatment, fluid management, dietary adjustments, and medication regimens may contribute to managing symptom experiences through self-care practices. Patients’ familiarity with chronic symptom experiences might lead them to express symptom intensity passively [46].

Confirming relatively easy-to-identify physical symptoms can be different from emotional symptoms such as sleep and sexual issues, worry, irritability, and sadness, which may be challenging for healthcare providers to recognize. Research has confirmed that healthcare providers tend to perceive physical symptoms directly related to hemodialysis treatment more seriously than emotional symptoms compared to patients [39]. Particularly in Korean traditional customs, discussing sexual problems is considered difficult, and nurses may avoid discussing these issues during patient assessments [47]. Effective communication during education or counseling sessions for hemodialysis patients is crucial not only for accurately assessing physical symptoms but also for addressing emotional symptoms promptly. This underscores the need for consultations and assessments to take place in independent and private spaces. Moreover, hemodialysis nurses must possess effective communication skills to accurately understand the complex and varied symptoms and needs of hemodialysis patients. Regular training to enhance communication skills is essential for fostering effective communication in this context.

The quality of life (QOL) of hemodialysis patients averaged 2.93 points on a 5-point scale. The environmental domain scored the highest at 3.19 points, followed by the physical health domain, psychological health domain, social relationships domain, and overall QOL. These findings are consistent with previous research on the QOL of hemodialysis patients [13]. Subdomains that scored lower than the overall average included the psychological health domain, social relationships domain, and overall QOL. The lower score in the psychological health domain aligns with previous findings that hemodialysis patients often experience high levels of emotional symptoms [13]. The lower score in the social relationships domain is attributed to decreased social and daily life abilities due to frequent hemodialysis sessions and employment challenges. Individuals facing similar issues can benefit from support groups for comfort and encouragement, potentially changing their social attitudes [48]. Activating support groups such as self-help meetings can enhance social interaction and positively impact both physical health management and social well-being among hemodialysis patients. The lower overall QOL score reflects the chronic nature of the disease requiring lifelong dependence on dialysis machines to prolong life, with the threat of life-threatening consequences if treatment is discontinued [9]. These results imply that interventions to effectively improve the QOL of hemodialysis patients are necessary. Strategies and intervention programs should be developed based on various factors influencing QOL, and subsequent research is needed to evaluate their effectiveness.

Based on general characteristics, QOL showed significant differences based on religion, occupation, monthly family income, and comorbidities. It was found that higher monthly family income was associated with higher QOL in individuals who reported having religion and occupation, consistent with previous research indicating that QOL tends to be lower in those with comorbidities [25].

Religion is known to enhance subjective well-being, reduce depression, and decrease psychological distress, thus promoting both physical and mental health [49]. It fosters a positive attitude towards life’s meaning and goals, establishes self-identity, and is associated with higher QOL. These findings support the idea that religious involvement can contribute positively to maintaining and enhancing human health.

The quality of life of hemodialysis patients with low monthly family incomes tends to be lower, indicating a need for active intervention. This study also revealed that while many patients aspire to work, only 15.6% actually hold jobs due to reduced physical activity and regular dialysis treatments. National support policies are required to facilitate vocational rehabilitation through community collaboration for those seeking employment. Hemodialysis nurses should assist patients in maintaining employment by managing schedules such as changing days and times, and nighttime dialysis unit operations could also be a solution.

The majority of subjects (41.2%) rely on disability allowances provided by the government to sustain their livelihoods, given that their monthly family incomes are below KRW 1 million. As hemodialysis patients require lifelong treatment, their economic burden may become increasingly significant. Urgent government financial support is necessary, and expanding disability employment policies is essential to ensure steady incomes for hemodialysis patients. Hemodialysis nurses should also focus on collaborating with hospital social services teams to connect with welfare policies and social support funds. There was a significant difference in the quality of life depending on the presence or absence of comorbidities. The quality of life was higher in the absence of comorbidities. In a study targeting hemodialysis patients using the same tool [25], the quality of life differed depending on the presence or absence of comorbidities, supporting the results of this study. Therefore, it is thought that quality of life can be improved if prevention and education on complications and diseases that can occur during the dialysis period are provided. In addition, there is a need to standardize and implement work guidelines as there are some ambiguities in the work guidelines for hemodialysis in patients with comorbidities.

The analysis of the correlation between body image, symptom experiences, and quality of life in hemodialysis patients revealed that quality of life showed a positive correlation with body image and a negative correlation with symptom experiences. There was also a negative correlation between body image and symptom experiences. These findings are consistent with previous research indicating a negative correlation between quality of life and symptom experiences [13], as well as with results showing that better body image correlates with higher quality of life [50]. Improvements in symptom experiences among hemodialysis patients could lead to enhancements in body image and quality of life, underscoring the need for the accurate assessment of various symptoms and specialized nursing interventions based on assessment outcomes.

In this study, multiple regression analysis identified that the major factors influencing quality of life were body image, monthly family income (over KRW 3 million), and symptom experiences, which together explained 46% of the variance in quality of life. This aligns with previous studies highlighting the significant impact of symptom experiences and family income on the quality of life of hemodialysis patients [11,13], thereby supporting the findings of this study. Early education and preparedness regarding changes in body image among hemodialysis patients are crucial for enhancing their quality of life. Furthermore, developing and applying specialized assessment tools for evaluating body image in hemodialysis patients, as well as designing nursing intervention programs aimed at improving body image, are essential.

Policy measures for continuous economic support are required, such as increasing household income through vocational rehabilitation and job placement that takes physical limitations into account, and expanding employment policies for the disabled by considering the disability rating system for hemodialysis patients. Based on the research findings that symptom experiences mediate the impact of body image on quality of life in hemodialysis patients, nurses participating in patient care should maintain a continuous evaluation of the various physical and emotional symptoms experienced by hemodialysis patients. Symptom experience was identified as a significant factor influencing the quality of life of these patients, underscoring the need for systematic management that allows nurses to assess and integrate these factors into treatment processes.

This study provides foundational data for the development of nursing interventions aimed at enhancing the quality of life of hemodialysis patients by understanding the mediating effects of symptom experiences on the relationship between body image and quality of life. However, this study’s generalizability is limited by its reliance on data collected from a single hospital. Therefore, there is a need for expanded regional studies with increased sample sizes to validate these findings.

## 5. Conclusions

ESRD patients on hemodialysis suffer significant limitations related to disease and treatment modality. This study found that the key factors influencing quality of life in hemodialysis patients were body image, monthly family income (more than KRW 3 million), and symptom experiences, in that order. Additionally, it was confirmed that symptom experiences play a mediating role in the relationship between body image and quality of life in hemodialysis patients. This suggests that improvements in body image in hemodialysis patients may lead to a reduction in symptom experiences and, consequently, an improvement in quality of life. Therefore, in order to improve the quality of life of hemodialysis patients, it is necessary to provide an intervention program that can improve physical function by considering the subject’s physical activity level. In addition, there is a need to improve the level of education by providing various educational opportunities that take the characteristics of hemodialysis patients into account, and to develop a symptom management program that hemodialysis patients can control and manage on their own to help relieve symptoms.

## 6. Limitations

In this study, data were collected only from a single university hospital, so there are limitations to generalization, and repeated research with regional expansion and increased sample size is needed. Additionally, by using a self-administered questionnaire, the results may vary depending on the interpretation of the meaning of the questions. During the course of this study, it was noted that the body image measurement tool used had a large number of items and some ambiguous content, posing difficulties for participants in providing answers. Moreover, there were items not suitable for chronic disease patients such as hemodialysis patients, making accurate measurement challenging. Therefore, there is a need for the research and development of a suitable tool for assessing body image specifically tailored to hemodialysis patients. In addition, various symptoms and emotional complaints due to changes in body image were observed during the research process, so it is necessary to investigate the diagnosis of depression and whether antidepressants were taken, and qualitative research is needed to gain a deeper understanding and evaluation of the participants.

## Figures and Tables

**Figure 1 healthcare-12-01779-f001:**
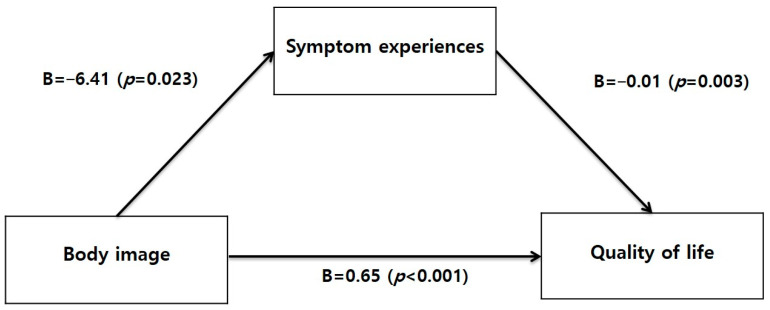
Mediating effect model of symptom experiences between body image and quality of life.

**Table 1 healthcare-12-01779-t001:** Participant characteristics and differences in quality of life (*n* = 153).

Variables	Categories	*n* (%)	Mean ± SD	t or F (*p*-Value)	Scheffé
Sex	Male	80 (52.3)	2.94 ± 0.46	−0.98 (0.331)	
	Female	73 (47.7)	3.02 ± 0.52		
Age (yr)	≤39	9 (5.8)	3.05 ± 0.63	0.53 (0.711)	
	40~49	11 ( 7.2)	2.85 ± 0.41		
	50~59	31 (20.3)	2.93 ± 0.43		
	60~69	57 (37.3)	2.98 ± 0.49		
	≥70	45 (29.4)	3.05 ± 0.52		
Marital status	Unmarried	30 (19.6)	2.82 ± 0.41	2.61 (0.077)	
	Married	102 (66.7)	3.04 ± 0.51		
	Others	21 (13.7)	2.91 ± 0.45		
Religion	Yes	72 (47.1)	3.07 ± 0.51	2.17 (0.031)	
	No	81 (52.9)	2.90 ± 0.46		
Education	≤Elementary school	37 (24.2)	2.85 ± 0.46	2.43 (0.068)	
	Middle school	30 (19.6)	2.95 ± 0.55		
	High school	54 (35.3)	2.99 ± 0.44		
	≥College	32 (20.9)	3.16 ± 0.52		
Occupation	Yes	24 (15.7)	3.29 ± 0.51	3.40 (0.001)	
	No	129 (84.3)	2.93 ± 0.47		
Monthly family income (KRW 10,000)	<100 ^a^	63 (41.2)	2.78 ± 0.45	10.62 (<0.001)	d,c > a
	100~199 ^b^	35 (22.9)	2.95 ± 042		
	200~299 ^c^	19 (12.4)	3.11 ± 0.51		
	≥300 ^d^	36 (23.5)	3.30 ± 0.45		
The person who helps the most	Spouse	69 (45.1)	3.05 ± 0.49	1.22 (0.305)	
	Parents	18 (11.8)	2.86 ± 0.40		
	Children	36 (23.5)	3.01 ± 0.52		
	Others	21 (13.7)	2.95 ± 0.52		
	None	9 ( 5.9)	2.73 ± 0.49		
Regular exercise	Yes	78 (51.0)	3.03 ± 0.48	1.35 (0.180)	
	No	75 (49.0)	2.93 ± 0.50		
Etiology CKD	Diabetes mellitus	89 (58.2)	2.85 ± 0.41	0.54 (0.613)	
	Hypertension	47 (30.6)	3.01 ± 0.61		
	Kidney stone	7 (4.8)	2.93 ± 0.43		
	Glomerulonephritis	6 (4.0)	2.91 ± 0.47		
	malignancy	4 (2.4)	2.95 ± 0.51		
Comorbidity *	Yes	94 (61.4)	2.87 ± 0.47	−3.71 (<0.001)	
	No	59 (38.6)	3.16 ± 0.48		
Frequency of hemodialysis (week)	Twice	37 (24.2)	3.06 ± 0.43	1.17 (0.244)	
	Three times	116 (75.8)	2.96 ± 0.51		
Hemodialysis periods(yr)	<2	40 (26.1)	3.07 ± 0.46	1.12 (0.344)	
	2~4	46 (30.1)	2.98 ± 0.51		
	4~9	44 (28.8)	2.98 ± 0.53		
	≥10	23 (15.0)	2.83 ± 0.44		
Moisture removal amount (kg)	<1	13 ( 8.5)	2.94 ± 0.30	1.15 (0.331)	
	1~1.9	26 (17.0)	3.09 ± 0.61		
	2~2.9	75 (49.0)	2.92 ± 0.47		
	≥3	39 (25.5)	3.05 ± 0.50		
Hospitalization experience in the past year	Yes	61 (39.9)	2.94 ± 0.49	−0.96 (0.336)	
	No	92 (60.1)	3.01 ± 0.49		

Comorbidity *: hypertension, diabetes, heart disease, arthritis, stroke, etc.

**Table 2 healthcare-12-01779-t002:** Degrees of body image, symptom experiences, and quality of life (*n* = 153).

Variables	Mean ± SD	Min	Max	Possible Range
Body image	2.88 ± 0.36	1.91	4.13	1–5
Emotional dimension	2.67 ± 0.50	1.63	4.33	1–5
Cognitive–behavioral dimension	3.02 ± 0.35	2.05	4.00	1–5
Symptom experiences	0.96 ± 0.79	0.00	4.47	0–5
Physical symptoms	0.94 ± 0.79	0.00	4.48	0–5
Emotional symptoms	1.03 ± 0.98	0.00	4.44	0–5
Quality of life	2.93 ± 0.46	2.04	4.31	1–5
Overall quality of life and general health	2.55 ± 0.86	1.00	5.00	1–5
Physical health domain	2.94 ± 0.46	1.71	4.00	1–5
Psychological domain	2.85 ± 0.52	1.67	4.33	1–5
Social relationships domain	2.66 ± 0.78	1.00	4.67	1–5
Environmental domain	3.19 ± 0.59	1.50	5.00	1–5

M = Mean; SD = Standard deviation.

**Table 3 healthcare-12-01779-t003:** Correlations among body image, symptom experiences, and quality of life (*n* = 153).

Variables	Body Image	Symptom Experiences	Quality of Life
	r (*p*)	r (*p*)	r (*p*)
Body image	1		
Symptom experiences	−0.18 (0.023)	1	
Quality of life	0.61 (<0.001)	−0.31 (<0.001)	11

**Table 4 healthcare-12-01779-t004:** Mediating effect of symptom experiences between body image and quality of life (*n* = 153).

	Step 1				Step 2			
	Symptom Experiences				Quality of Life			
	B	SE	*t*	*p*	B	SE	*t*	*p*
(Constant)	60.66	12.28	4.94	<0.001	1.25	0.28	4.39	<0.001
Religion * (yes)	1.40	2.79	0.50	0.617	0.08	0.06	1.33	0.185
Occupation * (yes)	5.48	4.03	1.36	0.176	0.16	0.09	1.84	0.067
Monthly family income (KRW 100~199)	0.57	3.62	0.156	0.876	0.10	0.08	1.29	0.200
Monthly family income (KRW 200~299)	0.80	4.52	0.18	0.861	0.19	0.10	1.98	0.049
Monthly family income (≥KRW 300)	−0.47	3.87	−0.12	0.903	0.26	0.08	3.12	0.002
Comorbidities (yes)	8.40	2.96	2.84	0.005	−0.06	0.07	−0.91	0.367
Body image	−6.41	4.13	−1.55	0.023	0.65	0.09	7.26	<0.001
Symptom experiences					−0.01	0.01	−3.00	0.003
F(*p*) R^2^ (adj R^2^) d (d_u_)	2.01 (0.057) 0.090 (0.045) 1.96 (1.80)				17.21 (<0.001) 0.49 (0.460) 2.05 (1.82)			

d (d_u_): Durbin–Watson’s autocorrelation coefficient (upper critical limit) * Dummy variable: Occupation (no = 0), monthly family income (<100 = 0), diseases besides renal failure (no = 0); SE = standard error.

**Table 5 healthcare-12-01779-t005:** Mediating effect of symptom experiences between body image and quality of life (*n* = 153).

	B	Boot SE	Boot 95%CI	
			LLCI	ULCI
Body image → Symptom experiences → Quality of life	0.044	0.024	0.006	0.100

B: unstandardized coefficients; SE: standard error; LLCI: lower level of confidence interval; ULCI: upper level of confidence interval.

## Data Availability

The data sets used and/or analyzed in the current study are available from the author upon reasonable request.

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
