# Peer review of "The Mediating Effects of Symptom Experiences on the Relationship between Body Image and Quality of Life among Hemodialysis Patients in a Single Center"

_healthcare, 2024, doi:10.3390/healthcare12171779_

Round 1

Reviewer 1 Report

Comments and Suggestions for Authors The authors analyzed the relationship between body image and quality of life in a group of hemodialysis patients. The study group was small and came from only one center, which the authors described as a tertiary center, which may mean that the study population was not representative of the general population of chronic dialysis patients in their country. The study is only cross-sectional and not prospective. Most of the parameters analyzed are subjective, and the number of patients who did not complete the questionnaires is surprisingly small, suggesting that they were carefully selected from a larger population in the center where they are treated, but neither the selection criteria nor the percentage of selected patients from the overall population were provided. number of people treated in this center. Such an association study, while exploring an interesting area of ​​knowledge, would certainly require a larger and more diverse population, and patients would need to be followed for longer periods of time to determine more clinically relevant and objective associations with hemodialysis treatment outcomes. Comments on the Quality of English Language

minor coments,  several typos and omissions need revision

Author Response

Thank you very much for the careful advice from reviewers. It has been revised as follows. The content corresponding to the revised contents and responses to the reviewer's comments are marked in red in the text.

Reviewer 2 Report

Comments and Suggestions for Authors

The manuscript presents a study on the mediating effects of symptom experiences on the relationship between body image and quality of life in hemodialysis patients.

Abstract: The phrase "collection period was from July 20, 2023 to August 11, 2023" should be rephrased for smoother reading, e.g., "Data were collected between July 20 and August 11, 2023."

The conclusion is somewhat vague, as it suggests the development of educational programs but doesn't specify what these programs should entail. The conclusion could be strengthened by offering concrete suggestions for clinical practice or future research.

The discussion part is long, but informative.

It would be beneficial to address any limitations in the data collection or analysis processes. For instance, were there any challenges in recruiting participants or collecting data that might have influenced the results?

Author Response

(The authors gave the same response as above.)

Reviewer 3 Report

Comments and Suggestions for Authors

The manuscript by Yang evaluated the association between symptom experiences, body image, and quality of life in 153 patients undergoing chronic hemodialysis. The study's main findings included associations between body image, family income, and symptom experience with quality of life. While the study addresses an important topic concerning chronic hemodialysis patients, we recommend addressing the following points before proceeding:

1)       The author used the following questionnaires: MBSRQ (The Multidimensional Body-Self Relations Questionnaire), DSI (Dialysis Symptom Index) for measuring symptom experience, and the World Health Organization Quality of Life Scale (WHOQOL-BREF). Were all these questionnaires validated in the Korean population? Additionally, for hemodialysis patients, a specific questionnaire, the short-form Kidney Disease Quality of Life (KDQOL), is usually used to assess quality of life. What motivated the author to use a different quality of life questionnaire instead of the KDQOL? How comparable are these two questionnaires?

2)       In Table 2, the etiology of chronic kidney disease (CKD) should also be evaluated. For example, the presence of diabetes mellitus, either as the cause of CKD or as a comorbidity, has a more pronounced impact on CKD burden compared to glomerulonephritis. It is well-known that microangiopathic complications (such as retinopathy and neuropathy) and macroangiopathic complications (such as heart failure, peripheral arterial vascular disease, and stroke/TIA) result in a lower quality of life. Additionally, please clarify the term “disease besides renal failure” in the same table.

3)       Therefore, the results should be adjusted for the etiology of CKD and comorbidities.

4)       What percentage of patients had a diagnosis of depression or were taking antidepressant medications? Was the statistical analysis adjusted for these variables?

5)       When evaluating family income, was the presence of health insurance also considered? Having health insurance may directly impact the quality of life.

6)       Please make amendments to the discussion section accordingly after reviewing the above comments.

7)       There is room for improvement in the manuscript’s English grammar. The Abstract section needs revision for clarity and comprehension.  

Comments on the Quality of English Language

There is room for improvement in the manuscript’s English grammar. The Abstract section needs revision for clarity and comprehension.

Author Response

(The authors gave the same response as above.)

Round 2

Reviewer 1 Report

Comments and Suggestions for Authors

I maintain my earlier opinion that this small, single-center, cross-sectional study conducted in a non-generalizable population of chronic dialysis patients treated in a referral center and based on subjective parameters provides very limited new results. Therefore, the study should be treated as more of a hypothesis-generating one. I agree with the authors that major research is needed in this area. However, in my opinion, this study has too many caveats to be published in a reputable journal. Comments on the Quality of English Language

minor edition

Author Response

(The authors gave the same response as above.)

Reviewer 3 Report

Comments and Suggestions for Authors

Kindly address the following comments:

1)      The presence of comorbidities was added to Table 1 and Table 4, but it was not properly discussed in the Discussion section.

2)      Similarly, the lack of information about how many patients had a diagnosis of depression or were taking antidepressant medications was not included in the Limitations section, as was previously noted.

Author Response

(The authors gave the same response as above.)
